# Sensing and Device Neighborhood-Based Slot Assignment Approach for the Internet of Things

**Mushtaq Khan [1], Rahim Khan [2,3], Nadir Shah [1], Abdullah Ghani [2], Samia Allaoua Chelloug [4,*], Wasif Nisar [1] and Jason Teo [2]**

1   Department of Computer Science, COMSATS University Islamabad, Islamabad 47040, Pakistan
2   Faculty of Computing and Informatics, University Malaysia Sabah, Sabah 88400, Malaysia
3   Department of Computer Science, Abdul Wali Khan University Mardan, Mardan 23200, Pakistan
4   Department of Information Technology, College of Computer and Information Sciences, Princess Nourah bint Abdulrahman University, P.O. Box 84428, Riyadh 11671, Saudi Arabia
*   Correspondence: sachelloug@pnu.edu.sa

**Abstract:** Concurrent communication constitutes one of the challenging issues associated with IoT networks, as it is highly likely that multiple devices may start communication simultaneously. This issue has become more complex as devices belonging to the IoT networks increasingly become mobile. To resolve this issue, various mechanisms have been reported in the literature. However, none of these approaches has considered the neighborhood information of a server module to resolve this issue. In this paper, a neighborhood-based smart slot allocation scheme for the IoT is presented where member devices are mobile. In this scheme, every CH or server module is bound to maintain two different types of slots, i.e., dedicated and reserved. Dedicated slots are assigned to every device on a First-Come-First-Serve (FCFS) basis, whereas reserved slots are assigned to the migrated devices. Additionally, as long as a device $C_i$ is located inside the server module's coverage area, it is required to use these dedicated slots. Simulation results verified that the proposed neighborhood-based slot allocation scheme performed better than existing approaches and considerably improved various performance metrics, such as 20% in lifetime, 27.8% in slot allocation, and 30.50% in slot waiting time.

**Keywords:** Internet of Things; communication; authentication; TDMA





## 1. Introduction

Recent developments in Micro-Electro Mechanical Systems have made Electro-Magnetic Sensing and the Internet of Things (IoT) constitute challenging research areas, which is due to the overwhelming characteristics of these technologies, such as a convergence capacity with numerous emerging technologies and a wide range of application areas [1]. The IoT is a collection of various devices, i.e., $C_i$ and $S_j$, which are known as "things" where $C_i$ and $S_j$ represent ordinary and server devices, respectively, such that $i = 1 \ldots n$ and $j = 1 \ldots m$, i.e., $n > m$. A thing is defined as any physical (preferably mechanical) entity or object with different embedded modules, such as sensing, processing, and communication [2,3]. Additionally, these devices should be smart enough to form an operational networking infrastructure and possibly without human intervention [4–6]. Generally, devices belonging to the operational IoT networks are either static or mobile, i.e., (i) Both $C_i$ and $S_j$ are static (ii) For $C_i$ and $S_j$, the former is static whereas later is mobile (iii) $C_i$ and $S_j$ are mobile. It is to be noted that the communication mechanisms are the same—that is, wireless communication protocols—and preferably designed for resource constraint networks [5,7]. Cluster-enabled networking infrastructures are reported in the literature to account for how a device $C_i$ with limited transmission power should be able to share its collected data with the intended destination module—that is, server $S_j$ in this case. In these networks, devices, $C_i$ becomes divided into two groups: (i) an ordinary device and a (ii) cluster head (CH). The former device is used for data collection purposes only, whereas the latter is utilized to

process and send it to the centralized module. Usually in these networking infrastructures, i.e., heterogeneous networks, multiple devices $C_i$ are eager to transmit captured data to a common receiver, that is, a CH or server module $S_j$, and they preferably want to do this simultaneously. Furthermore, these resource-limited devices $C_i$ are designed such that their transceiver modules can either perform the transmission of packets or the receiving, and they are not capable of performing both activities simultaneously. Therefore, the packet loss ratio and collision are among the common issues associated with these networking infrastructures. These issues become more severe if multiple devices $C_i$ (preferably more than three) commence a proper communication sessions with a shared destination device $S_j$, i.e., commence concurrent communication [8,9].

Although it is a convincing approach, collision and hidden terminal scenarios are among the most common issues, which are required to be sorted on a priority basis. Likewise, distributed authentication control [10] and combined authentication/association [11] were reported to resolve the aforementioned issues. However, excessive registration time, the ratio of the collided packets, and latency are the challenges linked with these schemes.

One of the most promising random access mechanisms is time division multiple access (TDMA), which was reported to enable the concurrent frame transmission of more devices $C_i$ to a shared destination device, i.e., a server, with a certain degree of reliability [12]. In this scheme, every server $S_j$ is belted to divide its active time frame into slots of equal sizes. Additionally, a reliable scheduling approach, i.e., a slot assignment, should be followed to resolve the conflicting scenarios among multiple source devices. Furthermore, every device $C_i$ is forced to transmit captured data values, preferably in its dedicated slot. A channel-enabled (multi) TDMA and (FDMA) frequency-domain-multiple-access-scheme-based hybrid mechanism was reported that was specifically designed for wearable devices, and its main motive was the realization of reliable and collision-free communication schedules for the IoT [13]. In addition to this mechanism, an enhanced version of the TDMA, which was based on the topological ordering and slot waiting time of member devices, was presented. Given that every device $C_i$ knows about the waiting time of neighboring devices, hence, it successfully reduces the collision ratio among multiple competing devices [14]. A hybrid slot allocation and scheduling approach that was based on TDMA and CSMA/CA was presented, which was particularly focused on the core issue—that is, the maximum possible utilization of the on-board battery. Apart from it, numerous modes of member devices were introduced, such as active, sleep, idle and wake-up periods, to further improve the performance of the resource constraint networks, such as the IoT and wireless sensor networks [15,16]. Likewise, a co-operative scheme—that is, one based on the TDMA and ad hoc MAC—was reported in the literature to ensure a reliable transmission of the data collected by devices deployed at locations where direct communication with a server module was not possible. The selection of these device (preferably relaying devices) was based on their transmission range, time slot availability, and probability of success [17]. These approaches have enabled simultaneous communication among multiple competing devices; however, hidden and exposed terminal scenarios, overhearing, bandwidth wastage, and excessive waiting time for a dedicated slot were some of the challenging issues. To the best of our knowledge, a neighborhood-information-enabled TDMA scheme (preferably for IoT networks where member devices are mobile) has not been considered in the literature. Therefore, the proposed communication model should be able to utilize neighborhood information to devise a collision-free communication schedule for multiple competing devices in the operational IoT network.

In this paper, a neighborhood-information-oriented slot allocation scheme is presented to enable the data transmission (preferably simultaneous) of multiple mobile devices, including a common destination module, i.e., a CH or server module. The proposed neighborhood-enabled TDMA scheme is a promising communication scheme for situations where more mobile devices $C_i$ are eager to begin a proper communication session with a shared destination device. It is to be noted that the mechanism should (i) be efficient in

terms of energy consumption (b) have a minimum possible collision ratio and (c) have minimum overheads. The main contributions are summarized as follows:

1.  A neighborhood-oriented slots allocation scheme for the IoTs networks where member devices are mobile was created;
2.  An algorithm to ensure the maximum possible utilization of both dedicated and reserved slots in the operational IoT network was utilized;
3.  An infrastructure-free scheme for event-based application areas was elaborated.

The rest of the manuscript is organized as follows: In the subsequent section, we have tried to provide a detailed review, preferably comprehensive, of the most relevant techniques. In Section 3, the motivation behind the selection of the problem to resolve in this paper is provided, which is followed by a detailed discussion of the scheme presented in this paper. Simulation results, preferably in both textual and graphical formats, are described in Section 5 of the manuscript. Lastly, we concluded the paper by providing comprehensive remarks.

## 2. Literature Review

In the literature, numerous approaches have been reported to enable simultaneous communication or packet transmissions of multiple (preferably more than two) devices $C_i$ using a common transmission medium [18,19]. A comprehensive review of those existing mechanisms, which are closely related to the work presented in this article, is presented. Frequency division multiple access (orthogonal-based) approaches were presented to ensure the concurrent packet transmissions of more devices $C_i$ using a common medium [20]. In these schemes, overall bandwidth is divided into smaller frequency bands (also know as sub-channelizations) that are assigned to the competing devices that are eager to initiate a proper communication session. Single carrier FDMA resolves this issue by utilizing the time domain and the frequency domain information. Likewise, a non-orthogonal multiple access (NOMA) mechanism is an ideal solution for the aforementioned problems, preferably with the IoT networks [21]. A NOMA is the most promising random access technique (preferably radio access), which is the current de facto standard for wireless communication as well; it provides numerous benefits that are desirable in wireless communication, such as minimum possible latency with preferably maximum reliability, connectivity, and spectrum efficiency. Generally, a NOMA approach is designed to serve multiple devices competing for the same resources—that is, the receiving device $S_j$ in this case, particularly in terms of time and space. A power-based NOMA is presented where different users are bounded to send data using different power levels while maintaining the same radio band [22,23]. Although, these methodology are very successful in resolving the issue of the parallel transmission linked with IoT networks, bandwidth wastage is a still an open challenge linked to these schemes.

Apart from these approaches, a contention-based multiple access scheme was reported to resolve the scenarios where multiple devices are eager to initiate a proper communication session, preferably with a common receiver. The scenario becomes more complex if these devices are forced to transmit data via a shared channel [24,25]. It is based on a four-way handshaking scheme to minimize the collision probability of packets transmitted by multiple competing devices. However, registration time overheads and packet collisions are among the common issues with these schemes. Centralized authentication control [26], distributed authentication control [10], and combined authentication/association [11] were designed and developed to resolve various issues associated with the contention-based multiple access scheme. Although, these mechanisms have successfully resolved numerous issues linked to the contention-based schemes, various issues were introduced, such as transmission delay and an average packet collision ratio (APDR). In addition to the contention-based scheme, effective handshaking approaches, i.e., clear-to-send (CTS) and request-to-send (RTS), were reported to address the problem of the hidden terminal scenario in operational resource constraint networks [27]. Although they resolved the hidden terminal scenario,

they generated another problem—that is, an exposed terminal scenario where a device is bounded to be in a waiting state until a current transmission session is ended.

In [28], a slotted CSMA/CA and TDMA-based approach was developed to ensure collision-free communication sessions, preferably in scenarios where multiple active devices are eager to transmit data to a common receiver over a shared medium in the IoT network. Furthermore, this approach is useful in resolving excessive registration issues as well. Every member device uses the CSMA/CA approach to broadcast authentication requests, whereas the TDMA is utilized for the association request messages. Likewise, Zhai et al. [29] proposed a contention-free multiple access scheme to ensure the schedule-based transmission of packets in both the time and frequency domains. Additionally, an automatic repeat request approach was adopted to further strengthen its efficiency and reliability. However, it was applicable to a particular IoT hardware plate-form, and its applicability is questionable in other application domains. A token-enabled adaptive approach (preferably distributed) was presented to address one of the challenging issues—that is, hidden terminal scenarios—with mobile ad hoc networks (MANET) and IoTs [30]. However, the simultaneous transmission of packets, end-to-end delay, and one-point failure are some of the common issues with this approach. Similarly, a distributed TDMA-enabled approach was presented to resolve bandwidth wastage issue in resource constraint networks where various slots are assigned according to a pre-defined mature scheduling policy [31]. The priority of member devices was utilized to ensure a proper time slot allocation. However, concurrent communication among all devices was not guaranteed, as slots were assigned according to the priority, and low priority devices may starve for a particular resource. A naive multi-channel scheme, which was a hybrid of the TDMA and FDMA, was proposed to enable and ensure the establishment of proper communication sessions among various wearable devices in the Internet of Medical Things (IoMT) [13]. Energy efficiency, as well as time and space complexity, are tightly coupled issues with this approach. In addition to it, topological orders were utilized to develop an enhanced TDMA approach [14] where every device has prior knowledge about its waiting time for its assigned time slot. However, slot waiting time directly is proportional to the network density, i.e., the waiting time is comparatively longer if devices are densely deployed. Likewise, a distributed TDMA-oriented slots allocation mechanism was proposed by Bhatia et al. [32] where slots were randomly assigned. However, synchronization is one of the common issues associated with this approach. Batta et al. [33,34] presented an enhanced version of the distributed TDMA scheduling scheme to improve latency in the IoTs networks. These approaches have neglected an important metric—that is, the device neighborhood—in developing TDMA approaches.

## 3. Motivation

Simultaneous communication is among the common issues associated with resource constraint networks, such as the IoT and wireless sensor networks. To resolve this issues, TDMA-enabled communication approaches, which were specifically designed for the resource-limited networks, were reported in the literature. These approaches have resolved the aforementioned issue, i.e., the simultaneous communication of multiple devices with a common destination device, preferably with available resources; however, existing approaches were either designed for a particular application area or were very complex. As per our understanding and knowledge, these benchmark approaches (i.e., either flat or hierarchical networks) for available slots are either assigned in a dedicated or in a preemptive nature to numerous requesting or member devices $C_i$ in an operational IoT network. However,these approaches are not realistic, as devices that usually belong to the IoT or WSNs are deployed randomly. Apart from it, the majority of the existing approaches are based on an unrealistic assumption, i.e., that member devices $C_i$ in an IoT should be similar; that is not always possible in these networks. Moreover, existing schemes do not have a clear policy for scenarios where a member device $C_i$ is eager to move into the coverage area of another server or CH $S_j$ module in the operational IoT networks. Therefore, a reliable TDMA-enabled schemes needs to be developed that has the capacity to resolve the aforementioned issues (preferably, the mobility of member devices $C_i$)

without changing the available technological infrastructure of the IoT networks. Generally, every member device $C_i$ and server $S_j$ has a specific neighborhood, as depicted in Figure 1, at a particular time interval, irrespective of the device mobility. If a uniform slot allocation policy (static or dynamic), as prescribed by the existing approaches, is utilized throughout the IoT network where member devices $C_i$ are allowed to move from one server to another, then it is high likely that a portion of device will be in a constant waiting state while highest priority devices $C_i$ are served first. However, if this slot allotment policy is refined according to the neighborhood of either a member device $C_i$ or a server $S_j$, then the above-mentioned problems will be resolved.

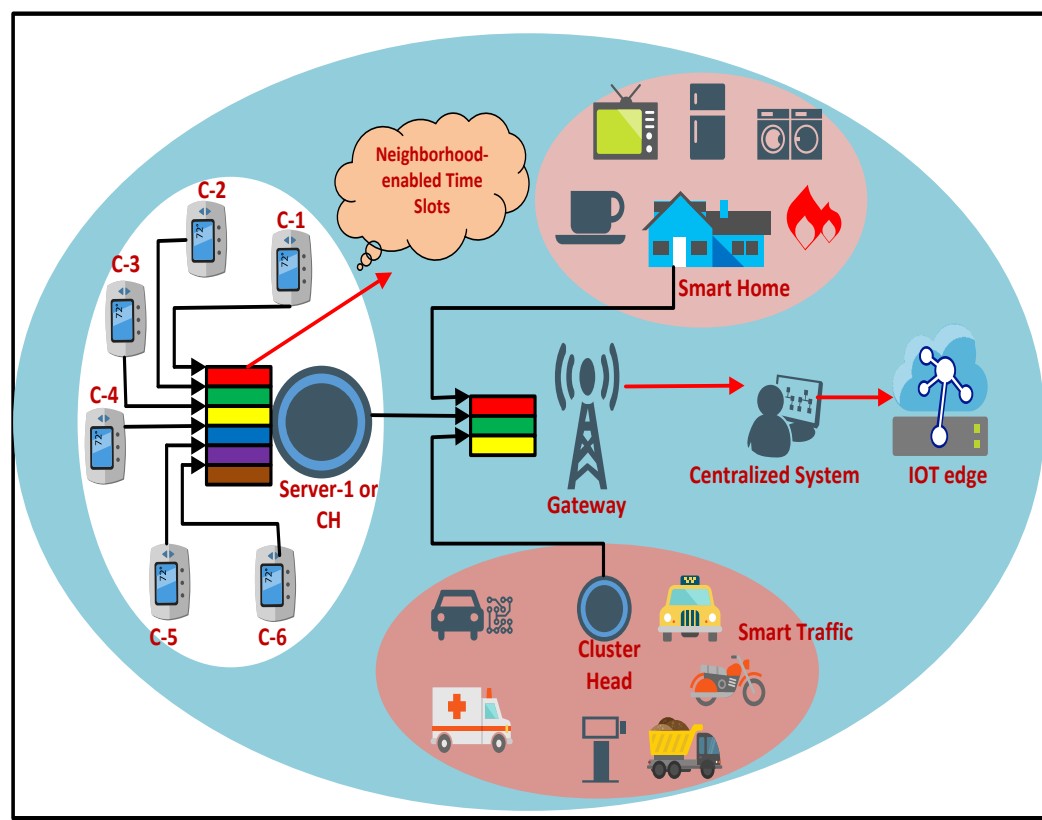

**Figure 1.** Neighborhood-Enabled TDMA approach.

## 4. Proposed Neighborhood-Enabled TDMA Approach

The neighborhood information of devices $C_i$ can play a vital role in developing an ideal TDMA-enabled communication approach for the networks in general and for the Internet of Things (IoT) in particular. If number of slots are equal to that of member devices $C_i$, which reside in closed proximity, then it is highly likely that every member device $C_i$ can transmit its collected data in its dedicated time slot. Furthermore, this refined mechanism has the capacity to resolve numerous other challenging issues, e.g., an average packet delivery ratio (APDR), starvation, collision, a minimum ratio of the latency and time interval (waiting) for a particular slot, and improvement of the lifespan of the respected IoT network. The proposed TDMA scheme is based on neighborhood information of the server $S_j$, where the number of time frames is divided into slots where the ratio of proportionality is direct to the respected devices $C_i$ that reside in the vicinity of the respective server.

### 4.1. Discovery of Neighbors in the IoT

In this phase, every server module $S_j$ is forced to broadcast a control packet, i.e., an $Msg_b$, where the value of the parameter join request is set to zero—that is, $Join_{Req} = 0$. This message, i.e., the $Msg_b$, is collected by those member devices $C_i$ that are placed in the

vicinity of the respective server $S_j$. The neighborhood or the vicinity of the device $C_i$ or server module $S_j$ is defined as presented in Equation (1).

$$
\begin{cases}
\forall_{a=0...n} \ C_a \in \sigma(S_b) \ \mathbf{iff} \dfrac{\sqrt{(C_{x_a}-S_{x_b})^2+(C_{y_a}-S_{y_b})^2}}{(x_a + y_b)} \ < \ \delta \\[4mm]
\exists_{a=0...n} \ C_a \in \sigma(S_b) \ \mathbf{iff} \dfrac{\sqrt{(C_{x_a}-S_{x_b})^2+(C_{y_a}-S_{y_b})^2}}{(x_a + y_b)} \ == \ \delta
\end{cases}
\tag{1}
$$

where variable S and C are used to describe the respective server and the devices in the networks, respectively.

In this equation, $\sigma$ and $\delta$ represent the neighborhood of the respective server module and threshold values (preferably distance-based), respectively. Devices $C_i$ with a distance value equal or less than the prescribed threshold value $\delta$ are potential candidate devices for joining the respective server $S_j$. If a device $C_i$ that fulfills the criteria (as described by the threshold value) is eager to become a member of the respective server module $S_j$, then, initially, it updates the received message $Msg_b$ such that the value of the request is to set to 1, i.e., $Join_{Req} = 1$, and it sends to the respective server $S_j$ as soon as the prescribed back-off timer is completed, which is defined as the time required by a particular member device $C_i$ to collect such messages from potential server modules. The back-off timer of a particular member device $C_i$ is obtained through Equation (2).

$$
Back-off\ Timer\ (C_i) \ = \ RAND(0\ to\ 100)\ microsec
\tag{2}
$$

Once the back-off timer is expired, then every device $C_i$ thoroughly examines the distance of numerous server modules and the overall strength of the received signal strength indicator ((RSSI) value. Generally, a device $C_i$ prefers to send a join request to that server $S_j$ that has a minimum distance and maximum value of the RSSI. For value of the RSSI, the following Equation (3) is used to compute the path loss ratio or function, which is an important measure to compute the RSSI value.

$$
P(d) = P(d_0) - 10n log \frac{d}{d_0} - X_\sigma
\tag{3}
$$

Once the path loss is computed, then the RSSI of the respective server module $S_j$ is calculated using Equation (4)

$$
RSSI(S_j) = P_t - P_{loss}(d)
\tag{4}
$$

Every device $C_i$ is forced to utilize these equations to compute RSSI values of the potential server modules $S_j$ (specifically those which are deployed in the vicinity) and send a modified version of the received message, or join request, to that server $S_j$ that has minimum possible distance and maximum value of the RSSI. This mechanism is repeatedly applied by every device $C_i$ in the operational IoT network to become a member of a particular server module $S_j$. In this way, every server module $S_j$ has a list of member devices. It is to be noted that, unlike in existing schemes where it is assumed that every server $S_j$ should have equal member device, this is not mandatory in the proposed TDMA-enabled approach.

### 4.2. TDMA-Enabled Communication Strategy

In this phase, every server module $S_j$ is bounded to develop a TDMA-based packet transmission mechanism specifically for its member devices $C_i$. In this scheme, the TDMA slot number has a direct proportionality ratio to that of active devices $C_i$, i.e., the time frame is separated into three slots if the respective server module has received join requests from three devices. It is to be noted that a server device $S_j$ is bounded to receive a message, i.e., a join request, from neighboring devices, and it has to ensure that this device is located in the designated coverage area. For example, if a server module $S_j$ has received join requests from ten (10) devices, then it is forced to create at least ten (preferably equal) slots by dividing

its sliding window (time). In addition, a sequential approach, that is, first-come-first-serve (FCFS), is utilized to assigned these time slots to various requesting member devices (as described in Equation (5)). To ensure a reliable communication infrastructure, these slots are assigned in a non-preemptive manner, where member devices $C_i$ are bounded to hold these slots for as long as is needed.

$$Max(C_i) = Waiting - time \tag{5}$$

Additionally, a server module $S_j$ should be able to only assign slots and membership to the devices of $C_i$ that have made a request, irrespective of the fact that other server modules $S_j$ have minimum requests. This is usually the case if devices are deployed randomly in the IoT networks. Additionally, there is the activity of the allocation of time slots, i.e., each $C_i$ is carried out by the respective server device $S_j$ shortly after the deployment process is completed. The proposed TDMA-enabled approach is a dedicated slots assignment approach that makes it highly likely that every server module $S_j$ has a set of devices $C_i$, which are different form other, because the deployment process is random. For example, assume that we have placed IoT networks that have a total of three servers $S_j$ such that $S_1$, $S_2$ and $S_3$ of every server $S_j$ have 9, 18, and 24 member devices $C_i$, respectively. By utilizing the proposed neighborhood-enabled TDMA approach, server $S_3$ is bounded to generate 20 time slots because it has twenty member devices $C_i$, and it should assign these slots using the FCFS approach for the requesting device $C_i$. Similarly, server modules $S_1$ & $S_2$ should generate eight (08) and fifteen (15) time slots, respectively, to facilitate their member devices $C_i$.

Additionally, the proposed neighborhood-enabled TDMA approach allows member devices $C_i$ to initiate a request for multiple time slots if needed. However, the assignment of these slots is subjected to various constraints, such as the respective member device (having been assigned) not having data to send, wherein this slot can probably be assigned to the requesting device in a preemptive fashion, i.e., if the respective device has data to send, then it is re-allocated immediately. However, it is not necessary that additional slots be adjacent. Furthermore, additional slots are allocated temporarily (preemptive fashion), i.e., if the respective device $C_i$ is interested in initiating proper communication in the upcoming session, it preferably does this in its own or a dedicated time slot. Therefore, the respective server module $S_j$ broadcasts a query message "MSG-Q" to all member devices $C_i$, preferably before the initialization of another session. This message has two important data fields that are represented by S, M, and L, where S means single slot (preferably a dedicated one), M means multiple slots, and L means that the device is about to leave the coverage area of the respective server module $S_j$ and join another, which is signified by J. Therefore, a device $C_i$, preferably a member, is required to deliver an updated version of the received message "MSG-Q" with a value of S or M if it wishes to interact during the next session. If and only if the relevant server module does not get any message in response to its inquiry message, it is presumed by default that the device is not interested in communication in the next session. It should be noted that the shift probability distribution $P_{shift}$, as indicated in the equation, is used to assign the extra free slots (6).

$$p_{shif}(C_i) = \frac{(1 - \gamma)\gamma^k}{(1 - \gamma^k)} \cdot \gamma^s \tag{6}$$

In this equation, variable s has values ranging from $1 \ldots k$, where $k$ depicts the available free slots number. It is interesting to note that the effective utilization of these slots is one of the main motivations behind this scheme. Additionally, in this scheme, the server module needs to ensure a minimum possible empty slot ratio, preferably zero, in every cycle or frame. Values range as $0 < \gamma < 1$ and $\gamma = C_{1,2,3,4\ldots n}\frac{-1}{k-1}$ such that $C_{1,2,3,4\ldots n}$ is used to describe a complete set of devices $C_i$, especially contestants or participants. The strategy adopted for the proper management and allocation of the multiple available slots is depicted in

Equation (6). It is important to note that a device allocates multiple slots **iff**, and multiple contestant devices $C_i$ are eager to allocate more slots for communication. Additionally, the process should follow either a sequential order or a prioritized fashion to handle such types of scenarios. However, this strategy is not applicable in situations where every device $C_i$ has data to transmit, and, thus, free slot availability is negligible. Additionally, the respective server $S_j$ will ignore such requests if free slots are not available.

In addition to this methodology, a server $S_j$ may be assigned additional responsibilities, i.e., it will forward the data values of those servers $S_{j+1}$ where it is not feasible for them to transmit packets directly. Therefore, these servers communicate with the nearest server through the dedicated slot that is reserved for this purpose. To implement this, a few slots, i.e., in every server $S_j$, are kept and allocated from **iff** to another server $S_{j+1}$ to make a request. Servers $S_j$, which do not have the capability to transmit directly, must follow the methodology presented by the following Equation (7), i.e., $\forall_{j=0...m} S_j \in \sigma(S_{j+1})$ for **iff**.

$$\frac{\sqrt{(S_{x_i} - BS_{x_j})^2 + (S_{y_i} - BS_{y_j})^2}}{(x_i + y_j)} < \delta \tag{7}$$

Additionally, time slots need to be kept for severs $S_j$, which are either placed on edges or far away from the respective sink module in an IoT network. These devices are deployed at a distance, i.e., far beyond the range of the communication module. Slots are reserved according to Equation (8), that is, $\exists_{j=0...m} S_j \in \sigma(S_{j+1})$ for **iff**.

$$\frac{\sqrt{(S_{x_i} - BS_{x_j})^2 + (S_{y_i} - BS_{y_j})^2}}{(x_i + y_j)} == \delta \tag{8}$$

In this equation, the variable $\delta$ is utilized to present the capability of the respective Xbee or radio module, which is preferably wireless, of the $S_j$. However, as opposed to the mechanism adopted for the allocation of temporary time slots to the respective devices $C_i$, slots reserved for other server modules are dedicated and are not assigned to another server $S_j$.

Let us consider a scenario that is depicted in Figure 1, where Server-1 $S_j$ has six member or neighboring devices $C_i$. Initially, Server-1 initiates the process by broadcasting a control message "$MSG_b$" to identify potential neighboring devices $C_i$. It is quite possible that those devices installed in close proximity or vicinity to the respective server $S_j$ will receive this message, i.e., $C_{1,2,3...6}$. These devices may also receive such messages from other server modules $S_j$ but are eager to join this particular server $S_j$, as it has a minimum possible distance and a maximum value of RSSI. Thus, these devices send a response message (preferably an updated one) to the respective server module $S_j$, wherein the join request value is set to one or true, i.e., "$Join - Request : 1$". It is to be noted that the response message is sent as soon as the back-off timer is expired. In situations when numerous devices $C_i$ are anxious to submit their packets simultaneously, random waiting time is provided to reduce the likelihood of packet collision. Server-1 $S_j$ splits the sliding window into equal slots according to the number of asking devices, in this case, 6 (six), when it gets these signals and once its back-off timer has elapsed, as illustrated in Figure 1.

### 4.3. Device Mobility in the Operational IoT Network

The proposed neighborhood-enabled TDMA approach supports the mobility of member device $C_i$ from the coverage area of one server $S_j$ to another server $S_{j+1}$. If a particular member device $C_i$ is interested in changing its position, then it must inform the respective server module $S_j$ by sending a response message with a value of L to the query message of the respective server module $S_j$. Once a server module $S_j$ receives query response message with value L, then it is added to the reserved slot class, which are assigned to those devices $C_i$ that have migrated from the coverage area of another server $S_j$. Additionally, these slots may be assigned to another requesting device if it has initiated a request for multiple slots.

During the movement, it is assumed that the member device is not allowed to send any message, as it is not currently a member of any cluster. Once this device $C_i$ reaches its intended destination, then it responds to the query message of that server $S_{j+1}$ with value "J", which has the minimum possible distance from its current position and the maximum value of the RSSI as described in Equation (9)

$$\forall_{i=0...n} C_i \in \sigma(S_j) \; \textbf{iff} \; \frac{\sqrt{(C_{x_i} - S_{x_{j+1}})^2 + (C_{y_i} - S_{y_{j+1}})^2}}{(x_i + y_{j+1})} <= \delta \qquad (9)$$

Server $S_{j+1}$ assigns a time slot to the requesting device $C_i$ if it is available and increases its number of member devices $C_i$. However, it is to be noted that dedicated time slots are not assigned to the mobile requesting device, as time slots are fixed and assigned via a preemptive policy. Furthermore, it high likely that the respective server module $S_j$ has an empty time slot, which is possible only if one its own member device $C_i$ is moved, wherein it is then assigned permanently. Additionally, this device $C_i$ (preferably the one which is moved) can communicate via a secondary server module $S_{j+2}$ if and only if a slot is available, and the optimized server module $S_{j+1}$ does not have any empty time at a particular time interval. This is possible only if the respective device $C_i$ responds to the query messages of both servers as depicted in Algorithms 1 and 2.

---

**Algorithm 1** Proposed Algorithm for mobile devices $C_i$

---

**Require:** Revised TDMA where Neighbor related Information is Utilized
**Ensure:** Allocation of Free Slots to Devices $C_i$
1:   $Member - Devices \leftarrow$ **Null**
2:   $NonMember - Devices \leftarrow$ **Null**
3:   $\triangle \leftarrow 30$ ms
4:   $j$ (Variable)$\leftarrow$ Server Module $S_j$
5:   $i$ (Variable)$\leftarrow C_i$
6:   **for** Server $S_j \in IoMTs$ **do**
7:     Create "$MSG_a$"
8:     Set value "$-Request - Join : 0$"
9:     Transmit "$MSG_a$"
10:  **endfor**
11:  **for** Device $C_i \in IoMT$ **do**
12:    **if** Value of (RSSI) of Current Server, i.e., $(S_{j+1...m} < S_j)$ & ($\triangle$:zero) **then**
13:      Update Message "$MSG_a$"
14:      Put value "$Request - Join : one$"
15:      Update Destination Module "$S_j$"
16:      Timer (Back-off) = random(0–10,000 ms)
17:      Transmit "$MSG_a$"
18:    **endif**
19:  **endfor**
20:  **for** Server $S_j \in IoMT$ **do**
21:    **if** $(Request - Join : one)$ **then**
22:      Place Device $C_i$ in $Member - Devices$
23:      Generate Free Slot for Device $C_i$
24:    **else**
25:      Place Device $C_i$ in $NonMember - Devices$
26:    **end if**
27:  **end for**
28:  **return** Time Slots based on Neighborhood Information

---

A neighborhood-enabled TDMA algorithm for mobile devices $C_i$ and servers (static) $S_j$ was developed as presented below. In this approach, a device $C_i$ is bounded to wait for a defined interval time, i.e., 30 ms in this case, to collect messages, and accept join requests

from numerous servers $S_j$, which are deployed in the respective vicinity. Additionally, every server $S_j$ is bounded to make sure that a fair slot allocation policy is adopted for every contestant device $C_i$.

The proposed TDMA-enabled algorithm that is presented below has built in support for the mobility of member devices $C_i$ from the coverage area of one server $S_j$ to another server $S_{j+1}$ in the operational IoT network.

---

**Algorithm 2** Proposed TDMA algorithm for mobile devices in the IoT Networks

---

**Require:** TDMA for Mobile Device $C_i$
**Ensure:** T Allocation of Free Slots to Devices (Mobile) $C_i$
 1: *Member − Devices* ← **Null**
 2: *NonMember − Devices* ← **Null**
 3: $\triangle$ ← 30 ms
 4: $j$ (Variable)← Server Module $S_j$
 5: $i$ (Variable)← $C_i$
 6: **for** Server $S_j \in IoMTs$ **do**
 7:    Generate "$MSG_Q$"
 8:    Set value "$S : 0, M : 0, L : 0, J : 0$"
 9:    Broad-Cast "$MSG_Q$"
10: **endfor**
11: **for** every member or neighbor $C_i \in S_j$ **do**
12:    **if** $C_i$ has data to send **then**
13:       Update "$MSG_Q$"
14:       Set value "$S : 1, M : 0, L : 0, J : 0$"
15:    **endif**
16:    **if** $C_i$ needs multiple slots **then**
17:       Update "$MSG_Q$"
18:       Set value "$S : 0, M : 1, L : 0, J : 0$"
19:    **endif**
20:    **if** $C_i$ is interested to move out **then**
21:       Update "$MSG_Q$"
22:       Set value "$S : o, M : 0, L : 1, J : 0$"
23:    **endif**
24:    **if** $C_i$ has moved and is interest to send data **then**
25:       Update "$MSG_Q$"
26:       Set value "$S : 0, M : 0, L : 0, J : 1$"
27:    **endif**
28:    Set Destination "$S_j$"
29:    Backoff Timer = rand(0–100 ms)
30:    Broad-Cast "$MSG_Q$"
31: **endfor**
32: **for** every $S_j \in IoT$ **do**
33:    **if** ($MSG_Q : m$) **then**
34:       Assign multiple slots if available
35:    **elseif** ($MSG_Q : J$) **then**
36:       Assign slot if available
37:    **elseif** ($MSG_Q : L$) **then**
38:       Slot is available
39:    **end if**
40: **end for**
41: **return** Time Slots based on Neighborhood Information for Mobile Devices

---

## 5. Simulation Results

In this section, a comprehensive and detailed analysis of the proposed neighborhood-enabled TDMA and existing approaches (such as a hybrid-approach TDMA [35], an opportunistic TDMA [36], a prioritized TDMA [28], a cooperative TDMA [37] and a cooperative

MAC [38]) is presented. These algorithms were implemented in OMNET++, which is an open source simulation software, and evaluated based on numerous performance metrics, such as average throughput, slot waiting time interval, empty slot utilization, and APDR. A random deployed methodology was adopted to generate the IoT network where an embedded delay is assumed in the transceiver modules of every device $C_i$. Additionally, various other parameters that were used in the simulation setup are shown in Table 1.

**Table 1.** IoT network parameters.

| Parameters | Values |
|---|---|
| Deployment Area of the IoT | 1200 m $\times$ 1200 m |
| Member Devices $C_i$ | 50–1000 |
| Server Modules $S_j$ | 20 |
| Preamble and Physical Headers $S_j$ | 15 us |
| Length of Beacon $S_j$ | 75 to 110 bytes |
| Back-off Time $S_j$ | random |
| IFS & Gaurd Time $S_j$ | 40 us |
| Slot $S_j$ | 70 us |
| SNR $p$ | 10 dB |
| Feedback Bits | 8 Bits |
| Energy ($E_i$) | 52,000 mAh |
| Residual Energy ($E_r$) | $E_i$–$E_c$ |
| Transmission Power ($P_{T_x}$) | 91.4 mW |
| Channel Delay ($Ch_{delay}$) | 15 ms |
| Receiving Power ($P_{R_x}$) | 59.1 mW |
| Power Consumption (IM) | 1.27 mW |
| Power Consumption (SM) | 15.4 µW |
| Transceiver Energy ($T_i$) | 1 mW |
| Transmission Range ($T_r$) | 500 m |
| Receiving Power Threshold ($RTS_n$) | 1024 bits |
| Packet Size ($P_{size}$) | 128 bytes |
| Distance Between Devices | 300 m |
| Sampling Interval | 10 s |
| Topologies | Static and Random |

*5.1. Empty Slots Utilization*

One of the key criteria used to assess the effectiveness of the TDMA-enabled scheme in the live IoT networks is the slot usage ratio. Usually, a TDMA-enabled method with the highest feasible slot utilization ratio is seen as a great option for the conventional networks in general and particularly in the IoT if and only if other performance metrics are not sacrificed. In terms of slot usage ratio, the suggested neighborhood-enabled TDMA technique was compared to several field-proven algorithms. Figure 2 unequivocally demonstrates the proposed TDMA-enabled systems' superior performance over that of their competitors. Even though the suggested technique facilitates device mobility, it still outperformed the current strategy in terms of slot usage ratio. By allocating several vacant slots to a respective member device as long as these were accessible, the suggested technique attained the highest possible slots utilization ratio over existing schemes. In addition, IoT networks are used in a variety of event-driven application fields (ideally automatically) to help people manage different crucial occurrences. Moreover, the suggested neighborhood-enabled TDMA system was created for the event detection application area, but it is equally applicable in other application areas as well. Figures 3 and 4 demonstrates that the suggested strategy successfully attained the highest feasible usage ratio of open slots. These findings were gathered under various conditions, such as when all member devices were active and when only a specific percentage were. The suggested strategy performed better than the benchmark approaches used in both cases.

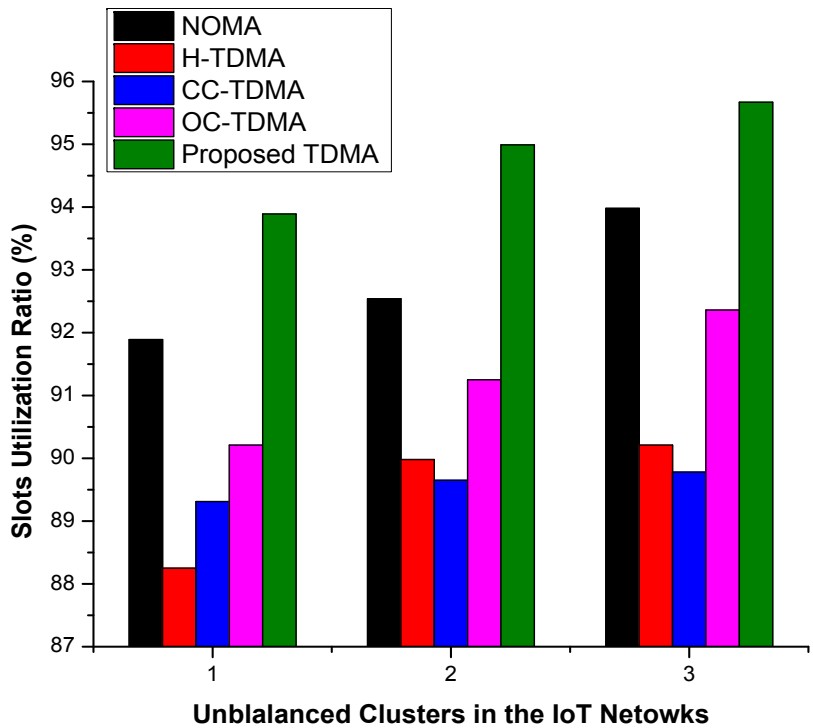

**Figure 2.** Empty slots utilization in the IoT network. Devices: 10.

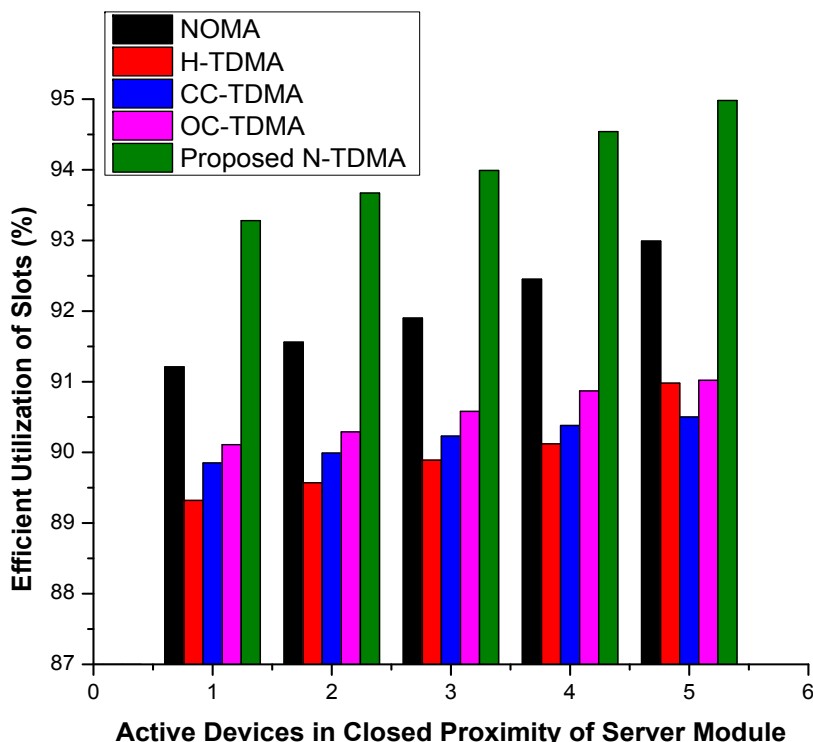

**Figure 3.** Empty slots utilization in the IoT network. Devices: 20.

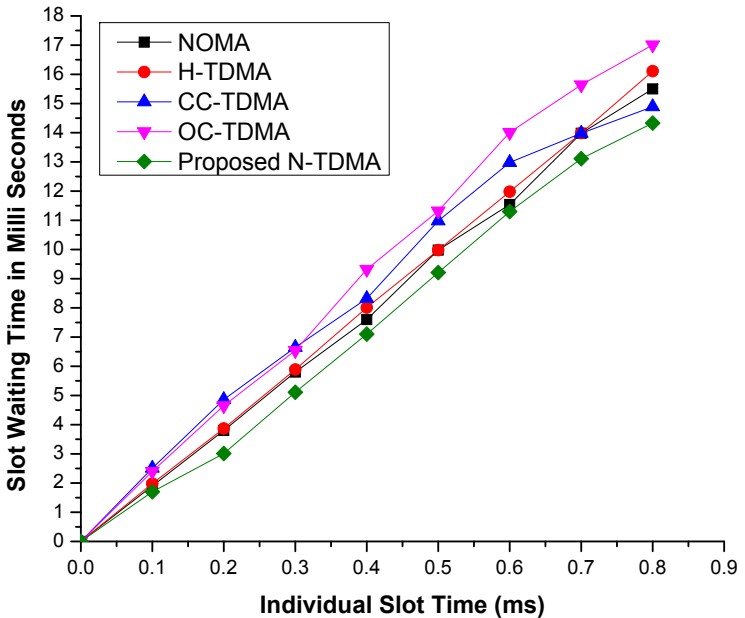

**Figure 4.** Slots Waiting time (ms) where member devices are less than or equal to ten (10).

### 5.2. Average Throughput Analysis

Average throughput is one of the most well-known measures used in the Internet of Things to assess how well communication methods function. It is defined as the quantity of sent packets or frames that are successfully received by the target device. Figure 5 illustrates a comparison of the performance of the suggested and current ways, in terms of average throughput, and it shows that the former mechanism was the best option for IoT networks, since the highest average throughput was possible at different network densities. Additionally, several networking typologies and a non-balanced clustering method, which is typical in IoT networks, were used to gather these data. Furthermore, we found that, unlike existing schemes, which have a direct link with these metrics, the performance of the proposed scheme was unaffected by changes in the number of member devices in a given cluster. This is similar to how previous systems function; the suggested approach's performance—in this case, average throughput—was unaffected by the proportion of active devices in a given cluster. Last but not least, the suggested solution supported member device movement, which did not impair the highlighted IoT network's overall performance.

### 5.3. Average Packet Loss Ratio (APLR)

The average packet loss ratio is the ratio of the number of transmitted packets to the number of packets that are not successfully received by the target device or are lost during communication, in this case, the base station, and it is inversely correlated with the average throughput of the IoT network. The suggested neighborhood-enabled TDMA scheme had a lower minimum APLR ratio than its opponent schemes when deployed in an IoT network according to a comparison of the proposed and current schemes (in terms of the average packet loss ratio, or APLR) provided in Figure 6. Additionally, in an unbalanced cluster environment seen in IoT networks, every server module $S_j$ is constrained to provide time slots that are identical to neighboring or member devices, which is impossible using current methods.

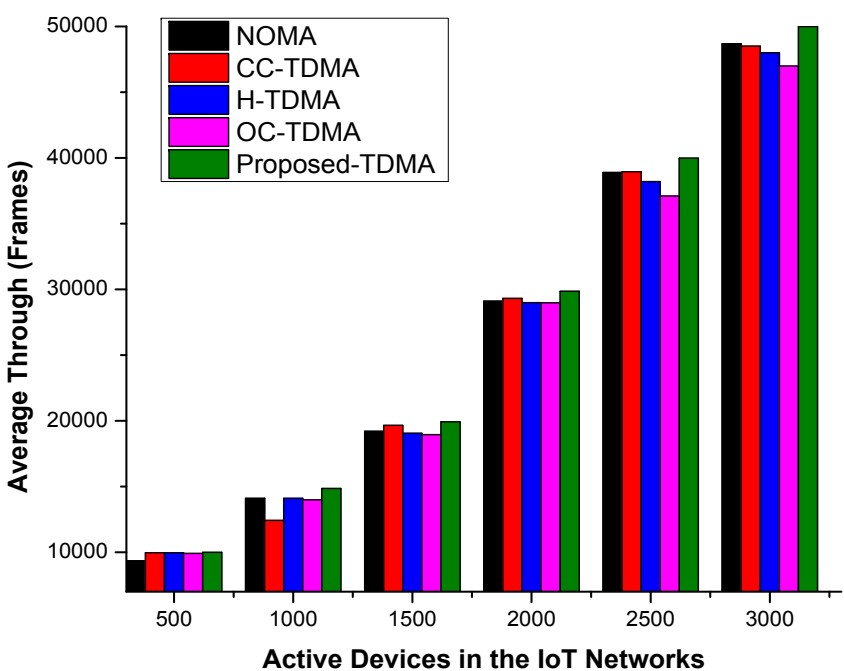

**Figure 5.** Evaluation of the proposed and existing methodologies in terms of throughput.

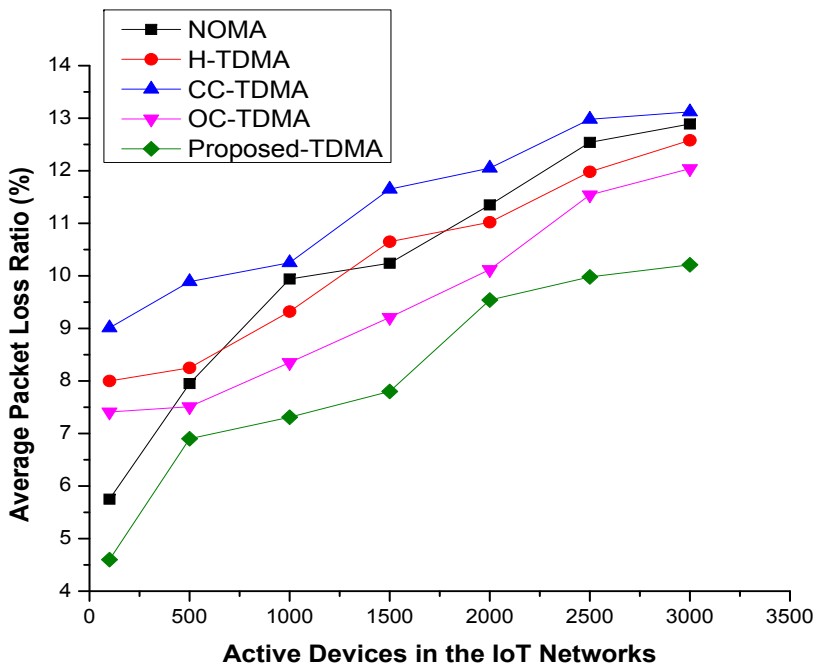

**Figure 6.** Evaluation of the proposed and existing methodologies in terms of APDR.

### 5.4. End-to-End Delay QoS Metric

End-to-end delay is assumed as a vital performance evaluation metric and is described as the total time taken by a packet transmitted from a sender to reach the destination module in the IoT network. Generally, this time is directly proportional to the number of slots, i.e., generated slots for communication, and active devices in the vicinity of a particular server module. As the proposed approach uses server's neighborhood information to develop a more informed slot development strategy, i.e., the TDMA, than the existing

schemes, this ultimately resulted in a minimum of possible time slots. However, these slots are enough to accommodate all active member devices, preferably those residing in the vicinity of server module, simultaneously. In Figure 7, it is clearly visible that that the proposed scheme successfully achieved the minimum possible delay ratio over the available state-of-the-art approaches in terms of different IoT networks.

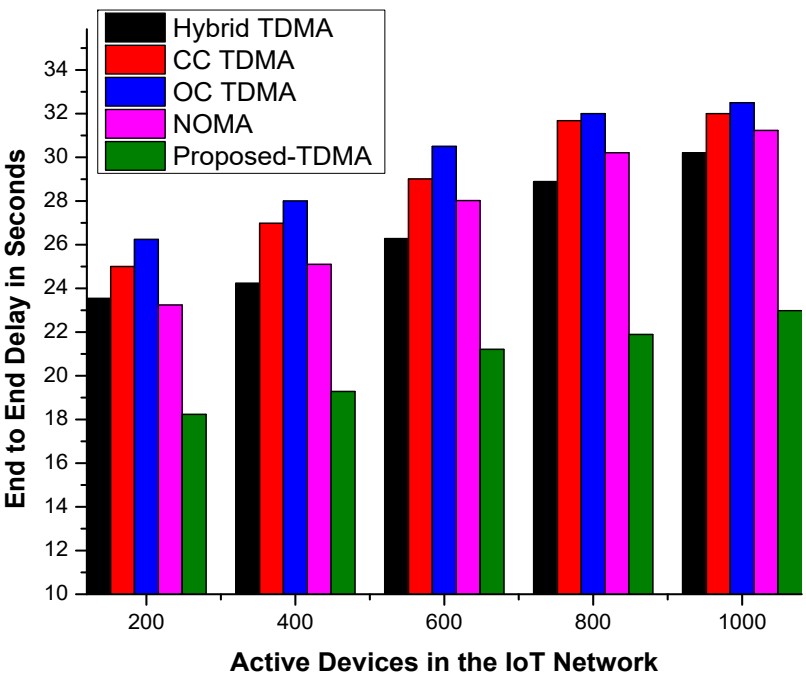

**Figure 7.** Evaluation of the proposed and existing methodologies in terms of end-to-end delay.

## 6. Conclusions and Future Work

Simultaneous or concurrent communication is among the challenging issues associated with the IoT networks that needs to be addressed on a priority basis. It is highly likely that majority of member devices may start communication with a common receiver simultaneously and even in the detection application. This issue becomes more complex if member devices belonging to the operational IoT network are mobile. To resolve this issue, various mechanisms have been reported in the literature, such as time division multiple access (TDMA), frequency division multiple access (FDMA), and non-orthogonal multiple access (NOMA), etc. However, none of these approaches have considered the neighborhood information of a server module or cluster head (CH), which can play a vital role in resolving the concurrent communication issue. In this paper, a neighborhood-based smart slot allocation scheme was presented for an IoT network where member devices were mobile, i.e., they could move from the coverage area of one server to another. In this scheme, every CH or server module was bounded to maintain two different types of slots, i.e., dedicated and reserved. Dedicated slots were assigned to every member device on a first-come-first-serve (FCFS) basis, whereas as reserved slots were assigned to those devices that migrated from the coverage area of another CH or server module. Additionally, a member device was bounded to utilize these dedicated slots, as long as it resided within the coverage area of the underlined server module. Simulation results verified that the proposed neighborhood-based slot allocation scheme performed better than existing approaches and considerably improved various performance metrics, such as 20% in lifetime, 27.8% in slot allocation, and 30.50% slot waiting time. In future, we are planning to present an extended version of the proposed neighborhood-enabled methodology for other infrastructures. Moreover, a multi-hop communication methodology could possibly be explored as well.

**Author Contributions:** Conceptualization, N.S. and R.K.; Methodology, M.K., R.K.; Investigation, W.N.; Resources, N.S., A.G.; Data Curation, W.N., A.G., J.T.; Writing—Original Draft Preparation, M.K., R.K.; Writing—Review & Editing, N.S., J.T.; Supervision, N.S., W.N.; Project Administration, R.K., A.G., J.T.; Funding Acquisition, S.A.C. All authors have read and agreed to the published version of the manuscript.

**Funding:** This work was supported by Princess Nourah bint Abdulrahman University Researchers Supporting Project number (PNURSP2023R239), Princess Nourah bint Abdulrahman University, Riyadh, Saudi Arabia.

**Informed Consent Statement:** This article does not contain any studies performed by the authors with human participants or animals.

**Acknowledgments:** We are grateful to the Princess Nourah bint Abdulrahman University Researchers Supporting Project number (PNURSP2023R239), Princess Nourah bint Abdulrahman University, Riyadh, Saudi Arabia, for supporting this project.

**Conflicts of Interest:** All authors declare that they have no conflict of interest.

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
