# Peer review of "Sensing and Device Neighborhood-Based Slot Assignment Approach for the Internet of Things"

_applsci, doi:10.3390/app13084682_

Round 1

Reviewer 1 Report

-          - Overview of the paper by sections is missing at the end of introductory section.

-          Please put the Figure 1 at appropriate place in the text or referred it appropriately. The Figure is in line 139, while first mention of the Figure is in line 283. Are you the author of the Figure? If not please provide reference.

-          Contributions are not clear. Comparison of the proposed method are compared to the existing solutions. Please introduce a section Discussion where you will clearly present contributions, it is of interest to summarize results in the table(s).

Author Response

We are very excited to have been given the opportunity to revise our manuscript titled “Sensing and Device Neighborhood based Slot Assignment Approach for the Internet of Things”. We have carefully considered the comments offered by the editor(s) and reviewer(s). We want to extend our appreciation for taking the time and efforts necessary to provide such insightful guidance and suggestions that led us to the improvement of paper contents. Very specific and short responses to the reviewers’ comments are provided below. Moreover, the changes are shown “blue” in the revised manuscript. 

Reviewer-1

Overview of the paper by sections is missing at the end of introductory section.

Response

We are thankful to the reviewer for providing these positive comments, which in facts improve both quality and readability of our paper. As suggested by the reviewers, organization section is provided in the revised paper as shown in Blue page. 3 for easy follow-up.

      Please put the Figure 1 at appropriate place in the text or referred it appropriately. The Figure is in line 139, while first mention of the Figure is in line 283. Are you the author of the Figure? If not please provide reference.

Response

      As suggested by the reviewer, we have put the Figure 1 at appropriate place in the text or referred it appropriately as shown in the revised manuscript and highlighted in blue colour for easy follow-up.

      Contributions are not clear. Comparison of the proposed method are compared to the existing solutions. Please introduce a section Discussion where you will clearly present contributions, it is of interest to summarize results in the table(s).

Response

      We are thankful to the reviewer for identifying these minor deficiencies. We have addressed these comments by providing main contributions of this paper in the second last paragraph of the introduction section especially in bullet format. These changes are highlighted in blue colour page. [3] in the revised manuscript. Additionally, it is important to note that we have provided a detailed comparative analysis of the proposed scheme with the available state of the art approaches preferably those related to the proposed solution in the simulation and result section.

Reviewer 2 Report

Authors propose a TDMA variation - a neighborhood-based slot allocation using slot assignment.

The paper is well organized, however it requires further improvement.

Please provide a table in section 2 where literature findings are given along with their characteristics and use-case applicability. Also, the objectives of the manuscript are not given with clarity. Please, elaborate further on this.

How do you differentiate with your previous work published in https://www.tandfonline.com/doi/full/10.1080/22797254.2021.1977717? What is  your further improvement? Please consider this in your graphs. H

Figures should be made wider and clearer.

Figure 1 is misleading. Authors should explain and describe further the architecture they propose. Why Cluster Head does not perform as and IoT Edge or the Gateway? if this is not the case, please explain further. Where is the cluster-head placed (e.g. microcontroller?)  How do you define the mobility? Mobile devices from traffic cluster or home cluster may move to the server-1 cluster? Is the computer or server located at the Centralized system responsible to execute the proposed TDMA algorithm? In this case, the gateway, the centralized system and the IoT Edge play all together the role of the IoT edge while the execution of the proposed TDMA algorithm is part of the edge computing mechanism. Please define accordingly.

Authors should also provide a table to compare the Proposed algorithm against the Existing Approaches. What is the complexity of the calculations? Could you compare the energy consumption per approach?

Does the proposed algorithm exhibit delays or add latency to the system response? What is the synchronization penalty if any, please provide more analysis and figure 4? 

Author Response

Response to reviewers

We are very excited to have been given the opportunity to revise our manuscript titled “Sensing and Device Neighborhood based Slot Assignment Approach for the Internet of Things”. We have carefully considered the comments offered by the editor(s) and reviewer(s). We want to extend our appreciation for taking the time and efforts necessary to provide such insightful guidance and suggestions that led us to the improvement of paper contents. Very specific and short responses to the reviewers’ comments are provided below. Moreover, the changes are shown “blue” in the revised manuscript. 

Reviewer-2

Authors propose a TDMA variation - a neighborhood-based slot allocation using slot assignment. The paper is well organized, however it requires further improvement.

Please provide a table in section 2 where literature findings are given along with their characteristics and use-case applicability. Also, the objectives of the manuscript are not given with clarity. Please, elaborate further on this.

Response

We are thankful to the reviewer for identifying these minor deficiencies. As suggested by the reviewers, findings and problem associated with the existing approaches are described and presented in the literature review section as highlighted in blue. Moreover, objectives of this paper have been presented very clearly in the introduction section as well as a separate motivation section is presented where major contribution or problem addressed in this paper is highlighted. Moreover, we have addressed these comments by providing main contributions of this paper in the second last paragraph of the introduction section especially in bullet format. These changes are highlighted in blue colour page. [3] in the revised manuscript. Additionally, it is important to note that we have provided a detailed comparative analysis of the proposed scheme with the available state of the art approaches preferably those related to the proposed solution in the simulation and result section. Finally, if the reviewer insist on providing these finding in tabular form, we are happy to do so.

How do you differentiate with your previous work published in https://www.tandfonline.com/doi/full/10.1080/22797254.2021.1977717? What is  your further improvement? Please consider this in your graphs.

Response

In previous work, which is already published, we have worked on static devices, i.e., both member devices and server modules were assumed to be static.  A common issue with this approach was the mobility of the wearable device which is quite common in today smart healthcare application.

Whereas, in this paper, we have considered that member devices are mobile, and they can change their positions and similarly the server module where these are registered. In this scheme, it is possible that a device may change it position and may be out of the coverage area of a particular server module and, thus, necessary to join another nearest server module for further communication. These were not possible with the exiting schemes.

Figures should be made wider and clearer.

Response

We believe that we have provide figures in clearer format, however, if the valuable reviewer insists on improving these further, then we are happy to do so.

Figure 1 is misleading. Authors should explain and describe further the architecture they propose. Why Cluster Head does not perform as and IoT Edge or the Gateway? if this is not the case, please explain further. Where is the cluster-head placed (e.g. microcontroller?)  How do you define the mobility? Mobile devices from traffic cluster or home cluster may move to the server-1 cluster? Is the computer or server located at the Centralized system responsible to execute the proposed TDMA algorithm? In this case, the gateway, the centralized system and the IoT Edge play all together the role of the IoT edge while the execution of the proposed TDMA algorithm is part of the edge computing mechanism. Please define accordingly.

Response

We are thankful to the reviewer for identifying these deficiencies. The proposed approach assumes that every member device has the capability to communicate directly with the server module, but due to the limited characteristic and random deployment nature of these devices, it is not always possible. Moreover, coverage area is another common issue associated with such type of assumption and scheme. Therefore, to address this, we have presented a multi sever architecture, where an IoT networks has more than one server, preferably located in such a way that every member device in the IoT network is deployed in the coverage area of at least one server module.

Mobility of the devices is defined as a device can change its position. For example, in smart healthcare system, every patient is equipped with wearable devices, and it is high likely that a patient may change its position, i.e. moving from one ward to another ward etc. Thus, in these scenarios, it is not always possible, a device will be able to communication with it designated server module.  Finally, it is important to note that these sever modules, which are static, are responsible to communicate with an edge where whole data is stored for onward processing.

Authors should also provide a table to compare the Proposed algorithm against the Existing Approaches. What is the complexity of the calculations? Could you compare the energy consumption per approach?

Response

We are thankful to the reviewer for providing these positive comments. We believe that a detailed comparative discussion along with supporting graphical represented in form of graphs have been provided in the revised manuscript. By providing this information in tabular form will be a duplication of information. However, if reviewers insist on providing this information in tabular form we are happy to do so.

Computational complexity is one of the most important measure for determining feasibility of a newly developed scheme. It is used to determine how much time an algorithm or procedure must consume if adopted to resolve an issue. Additionally, complexity analysis are used to determine the minimum possible requirement, i.e., preferably in terms of computation and space, of the electronic system, which if used should be able to perform according to the specifications. The complexity of the proposed scheme is O(n).

Finally, energy consumption of the proposed module along with existing state of the art approaches is presented in simulation section which is highlighted in blue for easy follow-up.

Does the proposed algorithm exhibit delays or add latency to the system response? What is the synchronization penalty if any, please provide more analysis and figure 4? 

Response

We believe that a detailed discussion on the latency performance metric has been provide in the revised paper which is highlighted in blue for easy follow-up. Additionally, this discussion is supported by the graphical results. Additionally, it is important to note that synchronization penalty of the proposed approach is far less than existing state of the art approaches as it allows only fist hop neighbours, i.e., devices reside in closed proximity, to communicate.

Reviewer 3 Report

In this paper, a neighborhood-based smart slot allocation scheme was presented for the IoT network where member devices are mobile, i.e., may move from coverage area of one server to another. In this scheme, every CH or server module is bounded to maintain two different types of slots i.e., dedicated and reserved. Dedicated slots are assigned to every member device in first come first serve (FCFS) basis whereas as reserved slots are assigned to those devices which are migrated from coverage area of another CH or server module. Additionally, a member device is bounded to utilized this dedicated slots as long as it resides within the coverage area of the underlined server module. Simulation results have verified that the proposed neighborhood-based slot allocation scheme performs better than existing approaches and considerably improves various performance metrics such as 20% in lifetime, 27.8% in slot allocation and 30.50% slot waiting time. The reviewer recommends major revision through:

1- Discussing how the proposed scheme may improves the quality indicators / quality of service for IoT;

2- The references section requires further improvements focusing on recent references published during last two years in IoT area;

3- Moderate English changes required.

Author Response

Response to reviewers

We are very excited to have been given the opportunity to revise our manuscript titled “Sensing and Device Neighborhood based Slot Assignment Approach for the Internet of Things”. We have carefully considered the comments offered by the editor(s) and reviewer(s). We want to extend our appreciation for taking the time and efforts necessary to provide such insightful guidance and suggestions that led us to the improvement of paper contents. Very specific and short responses to the reviewers’ comments are provided below. Moreover, the changes are shown “blue” in the revised manuscript. 

Reviewer-3

In this paper, a neighborhood-based smart slot allocation scheme was presented for the IoT network where member devices are mobile, i.e., may move from coverage area of one server to another. In this scheme, every CH or server module is bounded to maintain two different types of slots i.e., dedicated and reserved. Dedicated slots are assigned to every member device in first come first serve (FCFS) basis whereas as reserved slots are assigned to those devices which are migrated from coverage area of another CH or server module. Additionally, a member device is bounded to utilized this dedicated slots as long as it resides within the coverage area of the underlined server module. Simulation results have verified that the proposed neighborhood-based slot allocation scheme performs better than existing approaches and considerably improves various performance metrics such as 20% in lifetime, 27.8% in slot allocation and 30.50% slot waiting time. The reviewer recommends major revision through:

  • Discussing how the proposed scheme may improves the quality indicators / quality of service for IoT;

Response

            We are thankful to the reviewer for providing positive comments. In the results and discussion section, we have provided a detailed discussion on how the proposed approach could possibly resolve various issues, which are linked with the available state of the art approaches. Quality of service, specially from the perspective of the response time from sever module, is improved as devices connected to a particular server are limited and more importantly each has a dedicated slot for communication. Secondly, free slots are assigned to those devices which are in dire needs and have data to be shared with the intended server module. These are described in the result and discussion section as highlighted in blue for easy follow-up.

  • The references section requires further improvements focusing on recent references published during last two years in IoT area;

Response

As suggested by the reviewers, we have updated the reference section where reference have been provided from the recently published contents.

  • Moderate English changes required.

Response

We have proofread the whole paper as suggested by the valuable reviewer.

Round 2

Reviewer 1 Report

The authors addressed all my comments and I would like to propose the paper for publishing.

Author Response

We are very excited to have been given the opportunity to revise our manuscript titled “Sensing and Device Neighborhood based Slot Assignment Approach for the Internet of Things”. We have carefully considered the comments offered by the editor(s) and reviewer(s). We want to extend our appreciation for taking the time and efforts necessary to provide such insightful guidance and suggestions that led us to the improvement of paper contents. Very specific and short responses to the reviewers’ comments are provided below. Moreover, the changes are shown “blue” in the revised manuscript. 

Reviewer-1

The authors addressed all my comments, and I would like to propose the paper for publishing.

Response

We are thankful to the reviewer for providing valuable comments and in fact these comments have improved quality of the contents in the paper.

Reviewer 2 Report

Figures are still shown as stressed in. Please, reform them.

Efforts have been done to improve the manuscript, however it is almost the same with previously published paper from authors.

Comments have not been addressed adequately.

Author Response

Response to reviewers

We are very excited to have been given the opportunity to revise our manuscript titled “Sensing and Device Neighborhood based Slot Assignment Approach for the Internet of Things”. We have carefully considered the comments offered by the editor(s) and reviewer(s). We want to extend our appreciation for taking the time and efforts necessary to provide such insightful guidance and suggestions that led us to the improvement of paper contents. Very specific and short responses to the reviewers’ comments are provided below. Moreover, the changes are shown “blue” in the revised manuscript. 

Figures are still shown as stressed in. Please, reform them.

Response

As advised by the reviewer, we have included more high-quality figures with more improved visibility as compared to the previous version as shown in section 5 of the manuscript.

Efforts have been done to improve the manuscript; however, it is almost the same with previously published paper from authors.

Response

We believe that we have provided a detailed discussion of how the approach presented in this paper is different from the one which is already published. However, for further clarification, a comparative discussion is provided in the following table.

No.

Proposed TDMA Scheme

Published Scheme

1.

The proposed scheme is enhancement to our previous version where both device and server modules were assumed to be static, i.e., their position are fixed. However, in the scheme presented in this paper, a member device can change its position and could possibly move from the coverage area of one server module to another server module.

Both Member devices and Sever modules are static, i.e., these devices cannot change their respective position.

2.

Algorithm-I & Algorithm-II for allocation of time slots to mobile devices. Additionally, Algorithm-II described clearly how a device can move from coverage area of one server to another server module and what will happen to the existing slot. How this new device will be accommodated by the other server module. For this purpose, we have described this activity in subsection 4.3 as highlighted in blue colour for easy follow up.

Algorithm-1 for allocation of time slot to static devices

3.

Finally, In this scheme, we have reserved additional one or two slots with every server module which are allocated to server module deployed such that they are not able to communicate directly with the Centralized unit. Then, data of these servers are forwarded by the nearest server modules.

Missing in the published paper

4.

An example could be a patient admitted to a hospital and is equipped with wearable device to monitor his/her status. It is high likely that patient will need to visit other departments, i.e., X-rays and MRI, and it is not necessary that these facility will be available within the same building or within the coverage area of the current server module. The proposed scheme is an ideal solution for these situations.

 Not applicable with published paper.

Comments have not been addressed adequately.

Please provide a table in section 2 where literature findings are given along with their characteristics and use-case applicability. Also, the objectives of the manuscript are not given with clarity. Please, elaborate further on this.

Response

Initially, we have included a motivation section [Section-3] in the paper where a detailed discussion on why the proposed scheme is needed to be developed is presented in detail. In this section, we have identified various holes or problems especially with the concurrent communication in the literature and describe how these could be solved with the proposed approach.   

Secondly, as suggested by the reviewers, findings and problem associated with the existing approaches are described and presented in the literature review section as highlighted in blue [Page. 3 & 4 section 2].

Moreover, objectives of this paper have been presented very clearly in the introduction section as well as a separate motivation section is presented where major contribution or problem addressed in this paper is highlighted. Moreover, contributions of this paper are presented in the introduction section especially in bullet format Pages. [2 & 3] in the revised manuscript.

Additionally, it is important to note that we have provided a detailed comparative analysis of the proposed scheme with the available state of the art approaches preferably those related to the proposed solution in the simulation and result section. Finally, if the reviewer insists on providing these finding in tabular form like the one below, we are happy to do so, but we believe that this information is already provided in textual form.

Paper

Problem Addressed

Existing Solutions

Limitations

Proposed Solution

Design and

simulation of multi-channel v tdma for iot-based healthcare systems

Published: 2020

Reliable communication infrastructure

A multi-channel

variable TDMA

and FDMA (frequency domain

multiple access)

based scheduling

approach was presented to stream line a reliable

communication infrastructure for the

wearable devices

Complexity and energy efficiency are among the common issues associated with it.

Both issues addressed

Reviewer 3 Report

The authors fail to provide efficient revision seriously.

Author Response

Response to reviewers

We are very excited to have been given the opportunity to revise our manuscript titled “Sensing and Device Neighborhood based Slot Assignment Approach for the Internet of Things”. We have carefully considered the comments offered by the editor(s) and reviewer(s). We want to extend our appreciation for taking the time and efforts necessary to provide such insightful guidance and suggestions that led us to the improvement of paper contents. Very specific and short responses to the reviewers’ comments are provided below. Moreover, the changes are shown “blue” in the revised manuscript. 

Round-II Comments: The authors fail to provide efficient revision seriously.

As suggested by the valuable reviewer, we have provided a point-by-point response to the comments provided in Round-I and these comments are addressed and highlighted in blue colour in the revised manuscript for easy follow-up and review.

  • Discussing how the proposed scheme may improves the quality indicators / quality of service for IoT;

Response

            We are thankful to the reviewer for providing positive comments. We have provided detailed discussion on how the proposed scheme improves quality indicators/quality of service for the IoT in terms of different QoS evaluation metrics, i.e.,

  • End to End delay (both textual and graphical Page. 14 & 15, subsection 5.4)

The proposed scheme has the capacity to achieve minimum possible delay ratio as compared to the existing state of the art approaches which is highlighted in blue colour for easy follow-up and review.

  • Average Throughput Ratio (Page. 12 & 13) subsection 5.3

The proposed scheme has the capacity to achieve maximum possible average throughput ratio as compared to the existing state of the art approaches which is highlighted in blue colour for easy follow-up and review.

  • Average Packets Loss Ratio (APLR) (Page. 12 & 13) subsection 5.2.

The proposed scheme has the capacity to achieve minimum possible Average Packet Loss Ratio as compared to the existing state of the art approaches which is highlighted in blue colour for easy follow-up and review

  • Additionally, maximum utilization of the available slots is required as media is one of important resource in the IoT networks. The proposed scheme has shown consistent results in terms of both slot utilization and waiting time for a particular slot as well.
  • The references section requires further improvements focusing on recent references published during last two years in IoT area;

Response

As suggested by the reviewers, we have updated the reference section where reference have been provided from the recently published contents. Some of these references are given below.

  1. Lamya Alkhariji, Suparna De, Omer Rana, and Charith Perera. Semantics-based privacy by 482design for internet of things applications. Future Generation Computer Systems, 138:280–295, 483 2023. 484
  2. Cai Xu, Wei Zhao, Jinglong Zhao, Ziyu Guan, Xiangyu Song, and Jianxin Li. Uncertainty-aware 485 multiview deep learning for internet of things applications. IEEE Transactions on Industrial 486 Informatics, 19(2):1456–1466, 2022. 487
  3. Tuo Zhang, Lei Gao, Chaoyang He, Mi Zhang, Bhaskar Krishnamachari, and A Salman Aves- 488 timehr. Federated learning for the internet of things: applications, challenges, and opportunities. 489 IEEE Internet of Things Magazine, 5(1):24–29, 2022. 490
  4. Shahzad Khan, Waseem Iqbal, Abdul Waheed, Gulzar Mehmood, Shawal Khan, Mahdi Zareei, 505 and Rajesh Roshan Biswal. An efficient and secure revocation-enabled attribute-based access 506 control for E-health in smart society. Sensors, 22(1):336, 2022.
  5. Duc-Nghia Tran, Tu N Nguyen, Phung Cong Phi Khanh, and Duc-Tan Trana. An iot-based 494 design using accelerometers in animal behavior recognition systems. IEEE Sensors Journal, 2021. 495
  6. Muhammad Ali Naeem, Tu N Nguyen, Rashid Ali, Korhan Cengiz, Yahui Meng, and Tahir 496 Khurshaid. Hybrid cache management in iot-based named data networking. IEEE Internet of 497 Things Journal, 2021.
  • Moderate English changes required.

Response

We have proofread the whole paper as suggested by the valuable reviewer.

Round 3

Reviewer 2 Report

Authors have addressed the comments adequately.

Reviewer 3 Report

accept